

# A probe-based qRT-PCR method to profile immunological gene expression in blood of captive beluga whales (*Delphinapterus leucas*)

Ming-An Tsai[1,*], I-Hua Chen[2,*], Jiann-Hsiung Wang[2], Shih-Jen Chou[2], Tsung-Hsien Li[1], Ming-Yih Leu[1], Hsiao-Kuan Ho[3] and Wei Cheng Yang[2]

[1] Department of Biology, National Museum of Marine Biology and Aquarium, Pingtung, Taiwan
[2] College of Veterinary Medicine, National Chiayi University, Chiayi, Taiwan
[3] Department of Biology, Hi-Scene World Enterprise Co. Ltd., Pingtung, Taiwan
[*] These authors contributed equally to this work.

Corresponding author
Wei Cheng Yang, jackywc@gmail.com

## ABSTRACT

Cytokines are fundamental for a functioning immune system, and thus potentially serve as important indicators of animal health. Quantitation of mRNA using quantitative reverse transcription polymerase chain reaction (qRT-PCR) is an established immunological technique. It is particularly suitable for detecting the expression of proteins against which monoclonal antibodies are not available. In this study, we developed a probe-based quantitative gene expression assay for immunological assessment of captive beluga whales (*Delphinapterus leucas*) that is one of the most common cetacean species on display in aquariums worldwide. Six immunologically relevant genes (IL-2Rα, -4, -10, -12, TNFα, and IFNγ) were selected for analysis, and two validated housekeeping genes (PGK1 and RPL4) with stable expression were used as reference genes. Sixteen blood samples were obtained from four animals with different health conditions and stored in RNA*later*™ solution. These samples were used for RNA extraction followed by qRT-PCR analysis. Analysis of gene transcripts was performed by relative quantitation using the comparative Cq method with the integration of amplification efficiency and two reference genes. The expression levels of each gene in the samples from clinically healthy animals were normally distributed. Transcript outliers for IL-2Rα, IL-4, IL-12, TNFα, and IFNγ were noticed in four samples collected from two clinically unhealthy animals. This assay has the potential to identify immune system deviation from normal state, which is caused by health problems. Furthermore, knowing the immune status of captive cetaceans could help both trainers and veterinarians in implementing preventive approaches prior to disease onset.

## INTRODUCTION

In recent years, there has been rapid development in the field of cetacean immunology, resulting in new methods to prevent and treat infectious diseases in captive animals being used in education programs and naval defense. Because the free-ranging cetaceans may serve

as ideal sentinels of ecosystem health, efforts to develop reliable and relevant immunological techniques to address specific aspects of health and disease have increased (*Beineke et al., 2004*). Although cetacean species share many key immunological components with lab animals and humans, managing and evaluating the health of cetacean species remain a challenge (*Sitt et al., 2008*). Leukocyte transcriptional biomarkers such as cytokine genes have potential to assist in cetacean health assessment because of the broad scope of their functions and association with diseases known in human and veterinary medicine (*Chaussabel, 2015*). Cytokines are important in regulating the initiation, maintenance, and amplification of the immune response. Therefore, monitoring and evaluating a set of cytokines expressed within a certain microenvironment can be a diagnostic tool for characterizing immune responses to foreign antigens and vaccines (*De Jager et al., 2009*). It can also identify perturbations of the immune system induced by environmental insults.

Cytokine gene transcripts from several cetacean species have been recently cloned, and their DNA sequences have been determined. Quantitative analyses of cetacean gene transcripts have been reported in beluga whales (*Delphinapterus leucas*), Pacific white-sided dolphins (*Lagenorhynchus obliquidens*), bottlenose dolphins (*Tursiops truncatus*), harbor porpoises (*Phocoena phocoena*), and killer whales (*Orcinus orca*) (*Beineke et al., 2007*; *Buckman et al., 2011*; *Mancia, Warr & Chapman, 2008*; *Müller et al., 2013*; *Sitt et al., 2010*; *Sitt et al., 2008*; *Sitt et al., 2016*). These studies were based on quantitative reverse transcription polymerase chain reaction (qRT-PCR) using SYBR Green and various different house-keeping genes (HKGs). These studies enabled better understanding of the relative health of free-ranging cetacean species and *in vivo* baseline levels of gene expression in captive populations. Because of the short half-lives of leukocyte biomarkers and their tight transcriptional control, identifying disease-specific or antigen-specific patterns of cytokine gene expression could facilitate animal health maintenance.

qRT-PCR is a sensitive method commonly used in both basic and diagnostic research to quantify mRNA levels, and to provide informative measures of blood leukocyte gene transcripts. However, only probe-based quantification methods offer minimum non-specific fluorescence and high sensitivity to detect a single gene transcript compared with other dye-based chemistries, enabling an accurate quantification of the amplified targets (*Vanysacker et al., 2014*). Therefore, probe-based qRT-PCR with validated reference genes is a preferred method for precisely quantifying mRNA abundance and detecting of small changes in gene expression (*Wong et al., 2015*). Here we developed a species-specific probe-based qRT-PCR assay to measure the differential expression of immunologically relevant genes in beluga blood. Our findings could serve as a foundation for using transcriptional biomarkers for diagnosing diseases and assessing immunological profiles in captive and free-ranging cetaceans.

## MATERIALS & METHODS

### Sample collection and preservation

The voluntary blood collection of captive beluga in the National Museum of Marine Biology and Aquarium was performed according to the international guidelines (*Ramirez, 1999*).

**Table 1  Sample description.**

| Animal | Sex | Age (years) | No. of healthy samples (collection date: year/month) | No. of unhealthy samples (collection date: year/month) | Notes |
|---|---|---|---|---|---|
| A | Female | ~17 | 2 (2012/12; 2014/01) | 2 (2013/01, 09) | Open wounds |
| B | Male | ~17 | 4 (2013/01, 02, 08, 09) | 0 | |
| C | Female | ~13 | 2 (2013/04, 12) | 2 (2013/03; 2014/05) | Vesicles on fluke, low serum iron level |
| D | Male | ~17 | 4 (2012/12; 2013/01, 05, 09) | 0 | |

The voluntary blood collection means the animal has been trained to present its fluke and allow blood collection when the trainer shows a specific hand signal. When the animal showed reluctance to the voluntary blood draw, the trainers stopped the blood draw procedure and did some other training activity. Animal protocol was reviewed and approved by the Council of Agriculture of Taiwan (Approval number 1020727724). Sixteen blood samples from four 13~17-year-old adult animals (four from each animal, Table 1) were obtained via venipuncture of the fluke on a monthly basis from 2012 to 2014. Within five minutes of 1–2 mL blood being collected, 500 µL of EDTA-anticoagulated whole blood was preserved by adding 1.3 mL RNA stabilization solution (RNA*later*$^{TM}$; Ambion, Applied Biosystems, Foster City, CA, USA). Samples were stored at −20 °C until analysis. Samples from clinically healthy animals were used for establishing baseline values.

## RNA extraction and cDNA synthesis

The RiboPure$^{TM}$-Blood Kit (Ambion) was used for total RNA isolation from blood samples according to the manufacturer's instructions. RNase inhibitor (RNA Armor$^{TM}$ Reagent; Protech, Taipei, Taiwan) was added to the RNA solutions to prevent RNA degradation. RNA integrity was routinely examined using denaturing gel electrophoresis. RNA concentration was measured using fluorometer assay (Qubit$^{TM}$ fluorometer with a Quant-iT$^{TM}$ RNA Assay Kit (Invitrogen, Carlsbad, CA, USA)). Genomic DNA (gDNA) wipeout solution (Qiagen, Valencia, CA, USA) was added in the RNA samples for gDNA removal, and gDNA contamination was confirmed by qPCR prior to adding reverse transcription reagents. For complementary DNA (cDNA) synthesis, 73–444 ng of RNA and reverse transcription kit (QuantiTect®, Qiagen) were used. Unused extracted RNA and cDNA were stored at −80 °C.

## Primer and probe design

The sequences of six immunologically relevant genes (IL-2Rα, IL-4, IL-10, IL-12, IFNγ, and TNFα) of cetaceans (bottlenose dolphins, *Tursiops truncatus*) were obtained from GenBank (Table 2). The overall aim was to measure the gene expression of the immune-related activities, including pro-inflammatory, Th1/Th2, T cell growth, and anti-inflammatory features. For the probe-based qRT-PCR assay, a web-based software (ProbeFinder, v.2.49; Roche, Pleasanton, CA, USA) was used for designing specific primers and corresponding probes (Universal ProbeLibrary, Roche) (Table 2). Primer specificity of the six genes was validated using PCR (Fast-Run Hotstart PCR kit; Protech) and electrophoresis. Two validated reference genes (PGK1 and RPL4) (*Chen et al., 2016*) were included for normalization.

**Table 2  Name, accession number, primer sequence, probe number, amplicon size, efficiency and $R^2$ of 6 immunologically relevant genes.**

| Gene name | Accession number | Primer sequence (5′–3′) | UPL probe number | Amplicon size (bp) | Threshold | Efficiency (%) ± SD | $R^2$ |
|---|---|---|---|---|---|---|---|
| IL-2Rα | XM_004313501 | F-TGAACCTTTGAAGAGAATTTACCA R-CTGAATCCCTGAATGCACTG | 112 | 72 | 0.015 | 98.99 ± 1.25 | 0.998 |
| IL-4 | NM_001280657.1 | F-GCATGGAGCTGCCTGTAGA R-TGCAGAAAGTTTCCTTCTCAGTT | 140 | 69 | 0.012 | 93.92 ± 1.83 | 0.995 |
| IL-10 | AB775207.1 | F-AAGCCCTGTCGGAGATGAT R-CACGTGCTCTTTGATGTTGG | 25 | 86 | 0.012 | 97.25 ± 3.61 | 0.995 |
| IL-12 | XM_004324402.1 | F-CAGAAGGAGCTCTTTTATGACGA R-CCATGTGGTACATCTTCAAGTCC | 98 | 71 | 0.015 | 93.95 ± 3.04 | 0.997 |
| TNF-α | NM_001280615.1 | F-CCAACTGGCTACTCCATCATC R-CGGGCTTGTTACTTGAGGTT | 106 | 76 | 0.012 | 94.44 ± 2.00 | 0.997 |
| IFN-γ | AB022044.2 | F-TTTTCAGCTATGCGTGATTTTG R-TGCATTAAAATATTCCTTTAGGTTTTG | 129 | 94 | 0.010 | 95.78 ± 2.83 | 0.997 |

## Quantitative PCR

This study was conducted according to the MIQE guidelines (*Bustin et al., 2009*). cDNA was analyzed by quantitative PCR using FastStart Essential DNA Probes Master (Roche) according to the manufacturer's protocol. Thermal cycling was conducted using the Eco machine (Illumina, San Diego, CA, USA) and the same conditions were used for all target genes: 95 °C for 10 min for polymerase activation, followed by 45 cycles of 95 °C for 10 s and 60 °C for 30 s for denaturation and annealing/elongation, respectively. All reactions including plate controls and blank controls were run in triplicate. Plate controls include identical reaction materials on every run. A stable quantification cycle (Cq) value from all plate controls allowed data from multiple plates to be consolidated into a single data set. Threshold value for each candidate gene was manually set (Table 2). Baseline values were assigned for all plates using the Eco Software V4.0 (Illumina). PCR amplification efficiency ($E$) was calculated as $E = (10^{(-1/\text{slope})} - 1) \times 100\%$, where slope is the gradient of a standard curve. A gene-specific $E$ for the following normalized value (NV) calculation was obtained from the average of at least three $E$ values for each gene.

## Data analysis

Analysis of qRT-PCR data was conducted using NVs modified from *Pfaffl, Horgan & Dempfle (2002)*: $\text{Log}_2 (E_T^{CqT}/\text{Geomean} (E_{R1}^{CqR1} \& E_{R2}^{CqR2}))$, where $E_T$, $E_{R1}$, and $E_{R2}$ are efficiencies of the target gene and the reference genes, and CqT, CqR1, and CqR2 are Cq values of the target gene and the reference genes. Lower NVs indicate higher target gene expression levels. Outliers are defined as values more than the third quartile + 1.5 × IQR or less than the first quartile–1.5 × IQR, where IQR is the interquartile range.

## RESULTS AND DISCUSSION

$E$ values of all the six candidate genes ranged from 93.92% to 98.99% with $R^2$ values of > 0.99 (Table 2); therefore all six genes were included in the analysis of NVs. Figure 1 illustrates variable NV levels in the six candidate genes from clinically healthy samples with
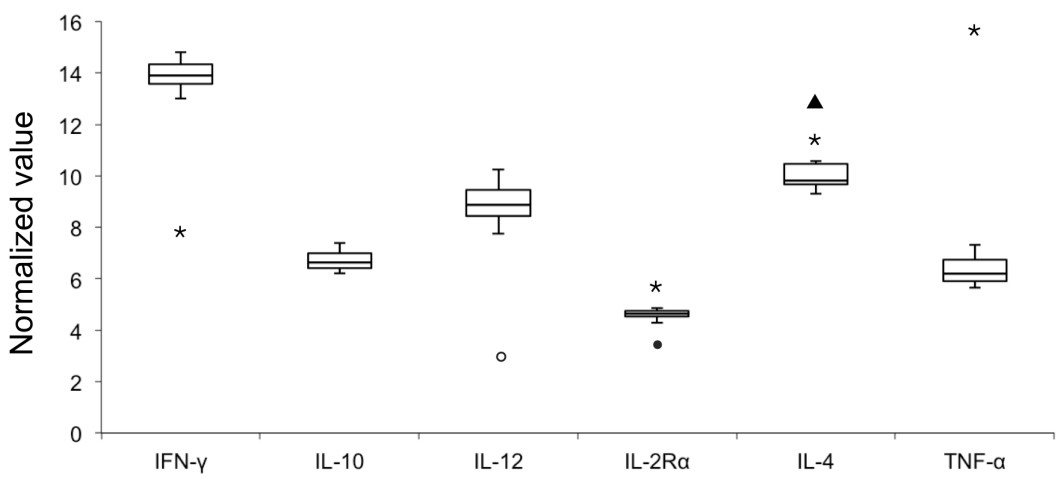

**Figure 1** Box plot of normalized value (NV) of 6 immunologically relevant genes for blood samples from clinically healthy (*n* = 12) and symptomatic (*n* = 4, outliers) belugas. The rectangle represents the second and third quartiles and the vertical line inside indicates the median value. The lower and upper quartiles are shown as horizontal lines either side of the rectangle. Open circle, sample A-I; closed circle, sample A-II; triangle, sample C-I; asterisk, sample C-II.

the lowest median NV (4.63) in IL-2Rα and the highest (13.89) in IFNγ. Gene transcript levels were used to establish three arbitrary categories: IL-2Rα (highly transcribed; median NV < 5), IL-10 and TNFα (moderately transcribed), and IFNγ, IL-4 and IL-12 (poorly transcribed; median NV > 8). Shapiro–Wilk test showed that NV data of each gene was normally distributed ($P > 0.05$). Stability in NVs during periods of health indicated the potential use in diagnostics by identifying outliers.

When pooling all samples (healthy and unhealthy) in one dataset (Table 2), NV outliers in IL-2Rα, IL-4, IL-12, IFNγ, and TNFα in four samples (A-I, A-II, C-I, and C-II) from two clinically unhealthy individuals were identified (Fig. 1). NV of IL-12 in sample A-I was 3.03 that is less than the first quartile–1.5 × IQR, demonstrating that the transcript level of IL-12 was elevated. NV of IL-2Rα in sample A-II was 3.45, that is less than the first quartile–1.5 × IQR, revealing that the transcript level of IL-2Rα was elevated. NV of IL-4 in sample C-I was 12.91 that is more than the third quartile + 1.5 × IQR, showing that the transcript level of IL-4 was decreased. NVs of IFNγ, IL-2Rα, IL-4, and TNFα in sample C-II were 7.91, 5.78, 11.45, and 15.50, revealing that the transcript level of IFNγ in this sample was elevated, whereas those of other genes were decreased.

In this study, we selected a panel of genes to cover a wide range of immunological events in beluga. The products of the selected genes are strong mediators of the immune system; they play a key role in the selection of immunological pathways and provide a link between innate and adaptive immune responses. IL-12 is a pro-inflammatory cytokine that induces proliferation and differentiation of T cells (*Hsieh et al., 1993*). TNFα is another pro-inflammatory cytokine that exerts cytotoxicity and induces cytokine secretion (*Clark, 2007*). It can also restrict the local spreading of infection. The α chain of IL-2 receptor is not expressed on resting T cells but only on activated T cells and is also called T cell activation (TAC) receptor (*Liao, Lin & Leonard, 2011*). IFNγ is produced by Th1 cells and shifts the

response toward a Th1 phenotype (*Schroder et al., 2004*). IL-4 suppresses the production of Th1 cells and is required for the production of IgE (*Sokol et al., 2008*). IL-10, an important immunoregulatory and anti-inflammatory cytokine (*Mosser & Zhang, 2008*), inhibits the synthesis of a number of cytokines involved in the inflammatory process, including IL-2, TNFα and IFNγ. It is also a promotor of Th2 response via the suppression of IL-12 synthesis. Altogether, the selected genes could reflect the complexity of immunological responses, and their products represent valuable immunological markers.

IL-12 in blood is mainly produced by neutrophils and monocytes in response to pathogens (bacteria, fungi, intracellular parasites and viruses) (*Trinchieri, 2003*). IL-2Rα expression on leukocytes (neutrophils, NK cells, and activated T helper and regulatory cells) has been reported as a potential non-specific marker of an activated immune system (*Zoldan et al., 2014*). Beluga A had open wounds on mandible and several traumas on head, fins, and trunk when sample A-I and A-II were collected, respectively (8-month difference between A-I and A-II collection date, Table 1). Although the blood work and behavior of beluga A did not show obvious signs of infection, the higher expression levels of IL-12 in A-I and IL-2Rα in A-II suggested that the immune system was activated as a result of wounds. Sample C-II was obtained when beluga C showed reluctance to the voluntary blood draw. Meanwhile, high fluke temperature was detected by infrared thermography, and vesicles were observed on the fluke of beluga C. Elevated expression levels of IFNγ and decreased levels of TNFα in C-II were suggested as normal responses to social stressors in the environment, as previously observed in a killer whale study (*Sitt et al., 2016*). However, the possibility of an immune response from a virus infection could not be ruled out because of the clinical findings of the fluke. Further virological investigations are ongoing. Sample C-I was taken when beluga C showed reluctance to the voluntary blood draw, and its blood work showed low serum iron level. Decreased expression levels of IL-4 in samples C-I and C-II were unexpected. IL-4 is known to play an important role in Th2 cell-mediated immunity, tissue repair, and homeostasis in human and experimental animals (*Gadani et al., 2012*). More studies are required to clarify the function of IL-4 in cetaceans.

Compared with solid organs, blood is a homogeneous tissue in which cellular composition can considerably vary depending on the location from where the sample is obtained. Without proper preservation, the copy number of individual mRNA transcripts in blood samples can change more than 1,000-fold during storage and transport (*Bowen et al., 2012*). Two commonly used methods for stabilizing blood RNA are the PAXgene Blood RNA vacutainer tube and RNA*later*™. These methods disrupt cells and precipitate RNA immediately upon homogenization by shaking the evacuated blood collection tubes. The collection tubes can then be stored frozen at −20 °C indefinitely without further processing. A previous study on the stability of RNA transcript from blood leukocytes using the above methods showed that both methods were suitable for use based on good quantity, integrity and purity of the isolated RNA (*Weber et al., 2010*). In this study, we used RNA*later*™ and smaller volume of blood (0.5 mL) to facilitate sample collection and transportation. Therefore, it was possible to adapt the methodology for serial sampling using small volumes of blood, which provides a temporal perspective transcriptome analysis.

It has been assumed that mRNA concentrations are the main determinant of the concentrations and activities of the corresponding proteins (*Vogel & Marcotte, 2012*). One should notice that recent studies showed that mRNA levels may only partially correlate with relative abundances of proteins (reviewed in *Vogel & Marcotte, 2012*) so protein abundances may or may not occur in proportion to their relative mRNA transcript abundances (*Ramakrishnan et al., 2009*). However, it has been shown that differentially expressed mRNAs correlate significantly better with their protein product than non-differentially expressed mRNAs (*Koussounadis et al., 2015*), providing optimism for the usefulness of mRNA data for biological discovery. Probe-based real-time assays with improved specificity are very useful in detecting low abundance cytokines for immunological research. It is particularly suitable for detecting the expression of proteins against which monoclonal antibodies are not available. The detection of cytokine mRNA using qRT-PCR has been suggested to be the only technique sensitive enough for reliable quantification *in vivo* (*Huggett et al., 2005*). The widely used SYBR Green assays in previous studies on cetacean gene transcripts have potential limitations such as primer-dimer formation, secondary structure formation by randomly binding to double stranded DNA, overestimate of target DNA, and higher inter-assay variation (*Vanysacker et al., 2014*). A fluorogenic probe-based approach with enhanced specificity was used in this study to prevent the limitations of SYBR Green assays. Amplification efficiency and reference gene selection are two important factors in gene transcript study using qPCR. The traditional NV is calculated using the $\Delta$Cq method (CqT$-$CqR). However, the traditional $\Delta Cq$ method can overestimate the error, and the calculation of the gene expression level requires correction when the amplification efficiency is not close to 100% (doubling of PCR products per cycle) (*Yuan, Wang & Stewart, 2008*). Moreover, using validated reference genes with stable gene transcript levels in varying experimental conditions can detect small perturbations with good sensitivity (*Dheda et al., 2005*). In this study, we determined the gene expression profiles by implementing rigorously calculated PCR amplification efficiency ($E$) and two validated reference genes. This is notably applicable to clinical sample with high variability and small changes in gene expression.

This study established a probe-based qRT-PCR assay for accurate and reliable detection and quantification of six immunologically relevant genes (IL-2R$\alpha$, IL-4, IL-10, IL-12, IFN$\gamma$, and TNF$\alpha$) and two validated reference genes (PGK1 and RPL4) in beluga. The real-time assay was successfully developed using a specific qRT-PCR protocol with the same chemistry and temperature profile, providing a simple and highly sensitive evaluation of normalized gene expression profiles. Preliminary data regarding the immune response of two clinically unhealthy beluga serves as a reference for future studies characterizing a range of health conditions of beluga. This tool for evaluating peripheral blood cytokine gene expression levels in cetaceans would facilitate research on the immune response of animals in the marine habitat in response to environmental insults, as well as the etiology of infectious diseases or stress.

## ACKNOWLEDGEMENTS

We thank the staff of the National Museum of Marine Biology and Aquarium for sample collection.

### Funding

This work was supported by Ministry of Science and Technology, Taiwan (grant number MOST 104-3113-E-002-012). The funders had no role in study design, data collection and analysis, decision to publish, or preparation of the manuscript.

### Grant Disclosures

The following grant information was disclosed by the authors:
Ministry of Science and Technology: MOST 104-3113-E-002-012.

### Competing Interests

Hsiao-Kuan Ho is an employee of Hi-Scene World Enterprise Co Ltd.

### Author Contributions

- Ming-An Tsai conceived and designed the experiments, analyzed the data, wrote the paper, reviewed drafts of the paper.
- I-Hua Chen and Wei Cheng Yang conceived and designed the experiments, performed the experiments, analyzed the data, wrote the paper, prepared figures and/or tables, reviewed drafts of the paper.
- Jiann-Hsiung Wang and Shih-Jen Chou conceived and designed the experiments, analyzed the data, contributed reagents/materials/analysis tools, wrote the paper, reviewed drafts of the paper.
- Tsung-Hsien Li, Ming-Yih Leu and Hsiao-Kuan Ho performed the experiments, contributed reagents/materials/analysis tools, reviewed drafts of the paper.

### Animal Ethics

The following information was supplied relating to ethical approvals (i.e., approving body and any reference numbers):

Animal protocol was reviewed and approved by the Council of Agriculture of Taiwan.

### Data Availability

The raw data has been supplied as a Supplementary File.

### Supplemental Information

Supplemental information for this article can be found online at http://dx.doi.org/10.7717/peerj.3840#supplemental-information.

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
