# Peer review of "A probe-based qRT-PCR method to profile immunological gene expression in blood of captive beluga whales (Delphinapterus leucas)"

_PeerJ, doi:10.7717/peerj.3840_

## Round 0.1 · original submission · Major Revisions

Your manuscript has to be deeply revised according to the reviewers comments. Please pay particular attention to the comments of reviewers 2 and 3, and be sure to address the ethical question.

·

Basic reporting

No comments

Experimental design

No comments

Validity of the findings

1. Although the present system shows some potentials to evaluate health conditions of captive beluga whales, it is better to combine other clinical pathological analysis results (e.g., complete blood count (CBC), serum chemistry panel) with the qPCR results to validate the system.
2. The limitations of the present study should be discussed. Changes on RNA level don't guaranty protein changes, especially when the gene changes a little.
3. The rationale selecting the 6 specific genes remain unclear. Both interleukin and respective receptor are crucial for downstream signaling pathway activation. Why some interleukin genes were selected, while a specific interleukin receptor was examined?

Additional comments

In the present study, the author investigated several inflammatory genes in captive beluga whales by quantitative RT-PCR. Generally, this is an interesting study and important to whale health monitoring in aquariums. Several other concerns need to be addressed before publication.
1. Animal backgrounds including place, age, sex and health conditions need to be included in a separated table, which is more straightforward to the readers.
2. The term of "A novel universal probe library quantitative reverse transcription polymerase chain reaction" in the title was confusing which needs to be revised. It should be careful to define the library as a universial library given its application area remain to be determined.

Reviewer 2 ·

Basic reporting

1. Line 65, ‘the short half-lives of leukocyte gene transcripts encoding surface and secreted messengers’ can be replaced with ‘the short half-lives of leukocyte biomarkers’.

2. Line 70, ‘mRNA level’ should be plural.

3. Include the full name of cDNA at Line 93 instead of Line 118.

4. Line 124, Cq is the abbreviation of crossing point.

5. Line 134, add ‘the’ before ‘target gene’.

6. Line 152, “NV of IL-12 in sample A-I was 3.03, demonstrating that the transcript level of IL-12 was elevated.” The authors need to say, “NV of IL-12 in sample A-I was 3.03 and that of the median was XXX, demonstrating that the transcript level of IL-12 was elevated.” Make similar changes for other genes.

7. In Figure 1, does the error bars represent standard deviation?

Experimental design

1. Immunological gene expression changes mainly in leukocytes after exposure, why the authors used whole blood cells instead of leukocytes? The authors need to validate the use of the whole blood cells or to use isolate the leukocytes.

2. There are commercially available ELISA kits to detect these cytokines, and ELISA doesn’t require blood cell separation, cell lysis or qRT-PCR. It is possible that those kits for human or mice cannot detect the cytokines from beluga. There could be one reason why people want use qRT-PCR instead of ELISA. Have the authors tested if the commercial kits work for beluga?

3. While it is clear that the qRT-PCR method is able to measure the expression levels of the genes, it fails to demonstrate its capability to evaluate the health status of the whales.

4. Line 138, ‘Elevated expression levels of IFNγ and decreased levels of TNFα in C-II were suggested as normal responses to social stressors in the environment, as previously observed in a killer whale study.’ How about other whales? How do the authors know they don’t suffer from social stress? Also, the authors took 4 samples from the same whale, why it is seen in only one sample.

Validity of the findings

1. Did the authors observe normalized value outliners in healthy samples, too? Please explain if there is.

2. The authors observed seven outliners but can only explain four of them, please explain why they thought it was enough. Also, if the elevated IL-12 expression in A-I whale suggested activated immune response, why they failed to observe its elevated expression in other 3 unhealthy samples. Same questions for IFN-γ, IL-2Rα, and TNF-α.

3. The authors used normalized value (NV) in this work. However, this term is not commonly used in qRT-PCR analysis. I suggest using ‘relative expression’. Why the authors define NV as Log2(ETCqT/Geomean (ER1CqR1 & ER2CqR2))?

4. What does ‘threshold value’ in Table 1 mean? How did the authors calculate it? People use the threshold cycle (Ct) in qRT-PCR analysis, but the ‘threshold’ here is clearly not Ct.

5. Line 70, it is real-time PCR but it doesn’t enable real-time gene expression measurement. The assay is off-line.

6. Line 144, why the authors use the median of NVs instead of the mean?

Additional comments

The paper described a method to quantify immunological gene expression in whole blood cells of beluga whales. The authors extracted mRNAs from the blood cells of healthy and unhealthy whales and compared the expression of six relevant genes. The authors attempted to find the correlations between the change of certain gene expression and health conditions. The paper is well organized, and easy to read. Probe-based qRT-PCR, such as Taqman, is a widely used method to evaluate gene expression, so the authors cannot claim it is a novel method. The authors need to avoid the use of 'novel' unless they can prove its novelty. Additionally, the authors need to improve the qRT-PCR analysis part and the discussion is weak.

1. Line 102 and Line 112, add the address information of the manufacturers.

2. Line 112, the authors need to put the data in supporting information.

3. Line 144, ‘Gene transcript levels were used to establish three arbitrary categories: IL-2Rα (highly transcribed; median NV < 5), IL-10 and TNFα (moderately transcribed), and IFNγ, IL-4 and IL-12 (lowly transcribed; (median NV > 8).’ The sentence can be removed.

4. Line 152, “NV of IL-12 in sample A-I was 3.03, demonstrating that the transcript level of IL-12 was elevated.” The authors need to say, “NV of IL-12 in sample A-I was 3.03 and that of the median was XXX, demonstrating that the transcript level of IL-12 was elevated.” Make similar changes for other genes.

5. SYBR Green was once widely used in qRT-PCR, but nowadays more and more people use probe-based Taqman. The Roche’s universal probelibrary is a new method, but it was not first introduced in this work.

6. Line 215, the most commonly used qRT-PCR analysis method is ΔΔCq, not ΔCq.

Reviewer 3 ·

Basic reporting

This was an interesting study on the immunology of captive belugas, providing preliminary data which may improve the health and welfare of captive cetaceans in the future.

Although the authors have done a very good job as (presumably) non-native English speakers with the language of this paper, I would still suggest that they request a native English speaker to proof-read the text throughout and to name this person in the Acknowledgements section, according to PeerJ/ICMJE criteria. Overall, the English is good, however, occasional sentences, such as on lines 33-34, lines 65-68, line 54, etc. could be rewritten to ensure their meaning is clearer.

In my opinion, the title is too long. I do not believe that anyone reading this paper needs the whole PCR procedure “spelling out”, surely this could be shortened by using qRT-PCR or RT-qPCR.

The Introduction provides a good background to the subject. Literature is current and relevant.

Perhaps it would be better to separate the Results and Discussion. This is the section order requested in the Instructions to Authors and it would appear quite straightforward to split this section.

Although Figure 1 is clear and relevant, I am not sure that “note” is the correct name for the symbol used for sample C-I, perhaps this could be changed to a more obvious and more commonly-used symbol to avoid confusion.

Experimental design

The research is within the scope of PeerJ.

Research question: clear. It is stated that immunological data on cetaceans is limited and research is needed.

Ethics: I have concerns about the “voluntary” aspect of the blood collection. While the authors state that international guidelines were followed (and I am sure that this is the case), these are not cited and I have been unable to find any such guidelines on the internet (I am sure they exist, but would like the authors should cite them directly). It is repeatedly stated in the results sections that beluga C showed reluctance to the “voluntary” blood draw and I feel that, when dealing with such sentient animals, this needs to be addressed and further explained by the authors.

M&M: While it is clear from the raw data that 16 blood samples were taken in total (i.e. 4 from each animal), and not 16 from each beluga, this is not so obvious in the M&M section. Suggest rewriting slightly. I would also suggest stating how much blood was taken from each animal, presumably at least 1ml was taken to allow for the exact measurement of the 500 µl EDTA-blood to be preserved.

Primer & probe design: Please state in the text that all gene accession numbers listed in Table 1 are from bottlenose dolphins.

Results: Were any of the results statistically significant? Particularly the differences between the sick and healthy animals. I realise, however, that the low number of samples probably makes a robust statistical analysis impossible.

Validity of the findings

Although 2 reference genes were used, they are not included in the results. Surely, if these genes were the control, they should at least be mentioned and not just included in the raw data.

Conclusions: While I would agree that studies of this kind are important in an attempt to improve the health of captive cetaceans, I would suggest to the authors that they tone down their final conclusions and at least add a statement that their results are based on only two clinically symptomatic beluga and so may not be representative of the general cetacean population.

Additional comments

Throughout:
Shouldn’t RNAlater be written with a trademark symbol?

Materials & Methods:
Line 83: The beluga are in captivity, not in “human care”, which sounds rather strange.
Line 100: Was gDNA contamination confirmed or excluded? Perhaps the absence of contamination was confirmed?
Line 101: This is the first mention of cDNA so I would add the definition (complementary DNA) here rather than on line 118.

Results & Discussion:
Line 146: I would suggest that the opposite of highly transcribed is poorly (not lowly) transcribed.
Lines 150-157: This text appears to repeat what is already shown in Figure 1. If there are any results in particular that the authors would like to highlight then I would do this here, but it doesn’t seem necessary to repeat all the information that is shown in the figure. Perhaps add the reference gene results here?
Line 186: Were any tests for possible viruses carried out on Beluga C? It would be interesting to know what was the cause of the symptoms, plus different pathogens could have different effects on the immune system.
Line 191: Perhaps add here that IL-4 is known to play an important role……….in humans, rather than just stating this as if the reference refers to cetacean research.

---

## Round 0.2 · Minor Revisions

Your paper needs some final improvements following the comments of one of Reviewer 3

·

Basic reporting

No comment

Experimental design

No comment

Validity of the findings

No comment

Additional comments

All concerns have been addressed.

Reviewer 3 ·

Basic reporting

No further comments.

Experimental design

Thank you for adding the reference to animal handling guidelines. Could you please explain your comment in the response letter “When the animal showed reluctance to the voluntary blood draw, the trainers suppose some thing happens”? Perhaps you mean the trainers stopped the blood draw procedure and did some other training activity? It would be good to add this to the Materials and Methods section of the text if this is the case.
The addition of Table 1 is a great help to the readers in understanding when blood samples were taken.

Validity of the findings

No further comments.

Additional comments

Lines 93-94: As the authors explained in their response letter, blood samples are taken from the fluke of the animal. As Beluga C was reported to have vesicles on the fluke perhaps it is important to mention that the area of blood draw may have influenced this animal’s “reluctance”. As such, I would suggest adding the following (or similar) to the blood sample method description “Sixteen blood samples……..were obtained via venipuncture of the fluke on a monthly basis…..”
Line 94: Please change to “Within five minutes of 1-2 mL blood being collected, 500µl…..”
Throughout: While RNAlater has been changed on line 96, ideally it should be corrected throughout the document.
Line 207: As the authors stated in their responses that another research group is currently working on virus investigation, it would be informative for readers to mention here that this research is ongoing
Table 1: please add “(years)” to the Age column
Figure legends: For clarity ,I would add that the outliers refer to clinically symptomatic animals, e.g. “…..blood samples from clinically healthy (n=12) and symptomatic (n=4, outliers) belugas.”

---

## Round 0.3 · accepted · Accept

Your revised manuscript can be accepted for publication in its present form. Congratulations!